# Soybean Seed Sugars: A Role in the Mechanism of Resistance to Charcoal Rot and Potential Use as Biomarkers in Selection

**DOI:** 10.3390/plants12020392

**Published:** 2023-01-14

**Authors:** Nacer Bellaloui, Alemu Mengistu, James R. Smith, Hamed K. Abbas, Cesare Accinelli, W. Thomas Shier

**Affiliations:** 1Crop Genetics Research Unit, USDA, Agricultural Research Service, 141 Experiment Station Road, Stoneville, MS 38776, USA; 2Crop Genetics Research Unit, USDA, Agricultural Research Service, Jackson, TN 38301, USA; 3Biological Control of Pests Research Unit, USDA, Agricultural Research Service, 59 Lee Road, Stoneville, MS 38776, USA; 4Department of Agricultural and Food Sciences, Alma Mater Studiorum, University of Bologna, Viale Fanin 44, 40127 Bologna, Italy; 5Department of Medicinal Chemistry, College of Pharmacy, University of Minnesota, 308 Harvard Street, SE, Minneapolis, MN 55455, USA

**Keywords:** soybean seed, seed composition, disease resistance, charcoal rot, seed sugars, seed quality, seed nutrition

## Abstract

Charcoal rot, caused by *Macrophomina phaseolina*, is a major soybean disease resulting in significant yield loss and poor seed quality. Currently, no resistant soybean cultivar is available in the market and resistance mechanisms to charcoal rot are unknown, although the disease is believed to infect plants from infected soil through the roots by unknown toxin-mediated mechanisms. The objective of this research was to investigate the association between seed sugars (sucrose, raffinose, stachyose, glucose, and fructose) and their role as biomarkers in the soybean defense mechanism in the moderately resistant (MR) and susceptible (S) genotypes to charcoal rot. Seven MR and six S genotypes were grown under irrigated (IR) and non-irrigated (NIR) conditions. A two-year field experiment was conducted in 2012 and 2013 at Jackson, TN, USA. The main findings in this research were that MR genotypes generally had the ability to maintain higher seed levels of sucrose, glucose, and fructose than did S genotypes. Conversely, susceptible genotypes showed a higher level of stachyose and lower levels of sucrose, glucose, and fructose. This was observed in 6 out of 7 MR genotypes and in 4 out of 6 S genotypes in 2012; and in 5 out of 7 MR genotypes and in 5 out of 6 S genotypes in 2013. The response of S genotypes with higher levels of stachyose and lower sucrose, glucose, and fructose, compared with those of MR genotypes, may indicate the possible role of these sugars in a defense mechanism against charcoal rot. It also indicates that nutrient pathways in MR genotypes allowed for a higher influx of nutritious sugars (sucrose, glucose, and fructose) than did S genotypes, suggesting these sugars as potential biomarkers for selecting MR soybean plants after harvest. This research provides new knowledge on seed sugars and helps in understanding the impact of charcoal rot on seed sugars in moderately resistant and susceptible genotypes.

## 1. Introduction

Soybean seeds are an important source for protein (40%), oil (20%), and carbohydrates (33%) [1,2,3]. Carbohydrates are present in soybean seeds as insoluble polysaccharides, which include pectin, cellulose, hemicellulose, and starch; and soluble carbohydrates, which include monosaccharides (glucose and fructose), disaccharides (sucrose), and raffinose family oligosaccharides (RFOs: mainly raffinose and stachyose) [4]. Total soluble carbohydrates (9–12% wt/wt) include 40–50 mg/g sucrose (C_12_H_22_O_11_), 20 mg/g raffinose (C_18_H_32_O_16_), and 35–45 mg/g stachyose (C_24_H_42_O_21_) [5], 2–3 mg/g glucose (C₆H₁₂O₆), 0.7–1 mg/g fructose (C_6_H_12_O_6_) [6]; 0.14–5.22 mg/g glucose; 0.00–2.93 mg/g fructose [7]. High levels of sucrose, glucose, and fructose are desirable, contributing to taste and flavor [2]. High raffinose and stachyose are undesirable, causing flatulence in monogastric animals [4], decreasing nutrient uptake [8], and reducing the nutritive value of seed and meal quality. On the other hand, raffinose and stachyose were reported to provide a protective mechanism against drought [9], cold [10], seed desiccation [11], reactive oxygen species [12], and carbohydrate partitioning during stress [9,13].

Charcoal rot is a major world-wide disease caused by the pathogen *Macrophomina phaseolina* that leads to severe loss in soybean yield and seed quality and sometimes leads to total yield loss (up to 100%) (Figure 1A–F), especially under drought/non-irrigation conditions (Figure 1D–E). Charcoal rot occurs in tropical and subtropic regions, as well as in the north central and southern regions of the United States [14,15]. *M. phaseolina* is a soil- and seed-borne fungus [16] that moves from roots to leaves, blocking vascular tissues [17] and decreasing nutrient uptake, water movement, xylem and phloem loading, and negatively impacts source–sink balance.

Strategies for charcoal rot control through agricultural practices, such as biological control and fungicide application, have all shown limited success [18,19]. Mechanisms of plant resistance to charcoal rot in plants are still unknown [15,20,21,22]. However, even though the mechanisms of resistance are unknown, the use of soybean genotypes with resistance to charcoal rot, developed through conventional soybean breeding, could be among the best and most sustainable strategies to protect high yield and seed quality. Currently, there are no resistant soybean cultivars available in the market [23,24,25]. However, in this study, soybean genotypes differing in resistance to charcoal rot were used, with the objective to investigate the possible role of sugars in the mechanism of soybean resistance and as biomarkers for soybean selection.

## 2. Materials and Methods

### 2.1. Growth Conditions

A two-year field experiment was conducted in 2012 and 2013 at the West Tennessee Research and Education Center at Jackson, TN, USA (35.65 N latitude, 89.51 W longitude). Thirteen moderately resistant and susceptible genotypes were used as shown in Table 1, which contains characterization information modified from Mengistu et al. (2018) [15]. The planting dates were 9 May in 2012 and 14 May in 2013. Field conditions were detailed elsewhere [15] and Bellaloui et al., 2021 [19]. Levels of *M. phaseolina* in the soil, expressed as colony forming units (CFUs) per gram of soil (CFU/g), ranged from 300 to 1000 CFUs [15]. To assess the severity of charcoal rot in soybean tissue, CFUs of *M. phaseolina* in the lower stem and roots at reproductive growth stage R7 [26] were estimated as previously described [15,19], with the data provided and presented in Table 2 (modified from Mengistu et al., 2018). Drip-irrigation was used to control irrigated and non-irrigated plots and continued to the end of R7 (physiological maturity). A buffer consisting of a four-rows was used between irrigated and non-irrigated plots to avoid water movement between plots [15].

### 2.2. Sucrose, Raffinose, and Stachyose Analysis

Mature seeds were analyzed for sucrose, raffinose, and stachyose concentrations using an AD7200 adiode array feed analyzer (Perten, Springfield, IL, USA) [3,27,28]. The calibration equation was established using Thermo Galactic Grams PLS IQ software (Perten, Springfield, IL, USA). The calibration equation was established using conventional laboratory protocols for sugar analysis using AOAC methods. Sucrose, raffinose, and stachyose were expressed as mg/g of dry weight. The seeds were immediately analyzed after harvesting in 2012 and 2013.

### 2.3. Glucose and Fructose Analysis

Mature seeds were analyzed for glucose using an enzymatic reaction of a Glucose (HK) Assay Kit, Product Code GAHK-20 (Sigma-Aldrich Co, St Louis, MO, USA), as detailed elsewhere [28]. The glucose concentration in seed samples was measured by a Beckman Coulter DU 800 spectrophotometer (Beckman Coulter Inc., Brea, CA, USA) by reading the samples at 340 nm. Fructose concentration was measured by an enzymatic reaction using a Fructose Assay Kit, Product Code FA-20 (Sigma-Aldrich Co., St. Louis, MO, USA) as described elsewhere [28]. Fructose concentration was measured by a Beckman Coulter DU 800 spectrophotometer (Beckman Coulter Inc., Brea, CA, USA) by reading the absorbance at 340 nm. The concentrations of glucose and fructose in seeds were expressed as mg/g dry weight. The seeds were immediately analyzed after harvesting in 2012 and 2013.

### 2.4. Experimental Design and Statistical Analysis

The experiment was a randomized complete block with a split–split plot with four replications, as described in detail by others [15,19]. Maturity groups (MG) were considered main plots, genotypes within MGs (IV and V) were sub-plots, and irrigation treatments (IR and NIR) were sub-sub-plots. PROC GLIMMIX (SAS, SAS Institute, Cary, NC, USA, 2002–2010) [29] was used for statistical analysis [19]. Random effects were the replicate and the replicate interactions with Year, Genotype, and IR [29]. Fixed effects were Year, Genotype, IR, and their interactions. Fisher’s Protected LSD test in SAS at the *p* ≤ 0.05 level of significance was used to compare means between all genotypes. Correlations between sugar variables were conducted by PROC CORR in SAS (SAS, SAS Institute, 2002–2010) as detailed by others [19].

## 3. Results

### 3.1. Analysis of Variance (ANOVA)

ANOVA showed that Year (Y) had a significant effect on sucrose, raffinose, and fructose (Table 3). Genotype (G) had a significant effect on sucrose, stachyose, glucose, and fructose. Glucose and fructose were significantly affected by Y*G interactions. Except for raffinose, Irrigation (IR) showed significant effects on all other sugars. Y*IR interactions were significant for stachyose and glucose. G*IR or Y*G*IR interactions had no significant effects on sugars (Table 3). Since Y*G and Y*IR significantly interacted for some sugars, results were presented by year and by irrigation.

### 3.2. Mean Values of Sugars

In 2012 and under IR, for sugar levels (mg/g) in moderately resistant (MR) genotypes, sucrose ranged from 29.23 to 39.83; raffinose from 5.48 to 6.29; stachyose from 25.28 to 42.10; glucose from 3.92 to 4.81; and fructose from 1.47 to 1.64 (Table 4). For susceptible genotypes (S) in 2012 under IR, sucrose ranged from 24.85 to 35.38; raffinose from 5.75 to 6.88; stachyose from 31.88 to 40.88; glucose from 4.48 to 5.62; and fructose from 1.15 to 1.40 (Table 4). In 2012 and under NIR, in moderately resistant (MR) genotypes, sucrose ranged from 28.00 to 36.38; raffinose from 5.07 to 6.44; stachyose from 26.35 to 45.43; glucose from 2.53 to 4.48; and fructose 1.03 to 1.35. In 2013, generally similar values were measured for sugar levels in harvested soybeans from plots under IR and NIR (Table 5), but there were notable exceptions. In 2013 under IR and NIR, except for raffinose under IR, the highest levels of sugars were recorded in the MR genotypes. In addition, MR genotypes had the ability to maintain higher levels of sucrose, glucose, and fructose under NIR conditions, where the disease was more severe. Susceptible genotypes showed higher levels of stachyose and lower levels of sucrose, glucose, and fructose under NIR. This was observed in 6 out of 7 MR genotypes and 4 out of 6 S genotypes in 2012; and in 5 out of 7 MR genotypes and 5 out of 6 S genotypes in 2013. Stachyose in S genotypes under NIR was highly significant compared to those under IR in 2012 and 2013 (Table 4 and Table 5). The noticeable degree of increase in stachyose in S genotypes under NIR was not observed in MR genotypes under NIR in 2012 and 2013. The same trend was observed for raffinose in S genotypes under NIR in 2012 and 2013, except for LS98 in 2012 and R01-581F in 2013. Sucrose, glucose, and fructose decreased dramatically in S genotypes under NIR in 2012 and 2013, compared with MR genotypes. Generally, the mean across all MR genotypes or the mean across all S genotypes showed similar patterns (Table 4 and Table 5). This was supported by the following results across years and genotypes; i.e., higher accumulation of raffinose and stachyose and lower sucrose, glucose, and fructose in susceptible genotypes, and higher sucrose, glucose, and fructose, and lower raffinose and stachyose in moderately resistant genotypes (Figure 2A–E).

### 3.3. Correlations between Sugars

Across genotypes in 2012 and 2013 under IR, inconsistent correlations between sugar components were observed. However, under NIR conditions, stachyose level was positively correlated with raffinose level and negatively correlated with sucrose, glucose, or fructose levels (Table 6). Sucrose was positively correlated with glucose or fructose levels. Glucose level was positively correlated with fructose and negatively correlated with raffinose or stachyose. Similar observations were noticed for fructose vs. raffinose and stachyose levels (Table 6).

## 4. Discussion

### 4.1. General Discussion

Mechanisms of resistance to charcoal rot are still unknown [16,30,31], and currently, there are no cultivars resistant to charcoal rot available to producers [15]. Therefore, understanding and exploring possible mechanisms involved in disease defense may be useful in future efforts to develop resistant cultivars. The lower levels of glucose, fructose, and sucrose in S genotypes, shown in the current study (Table 4 and Table 5), could be due to reduced xylem and phloem transport, and thus, nutrient uptake, caused by fungal interference associated with charcoal rot disease [32,33,34,35]. Reduced nutrient availability could result in reduced activity of the enzymes involved in synthesis of glucose, fructose, and sucrose, including enzymes such as those involved in sugar hydrolysis, sucrose synthesis, and production and turnover of raffinose and stachyose. The higher levels of glucose, fructose, and sucrose in MR genotypes and lower levels of these sugars in S genotypes under NIR conditions where charcoal rot is severe are consistent with the involvement of the charcoal rot pathogen in the inhibition of enzymes that produce mono- and di-saccharides and of the transport system.

Previous research showed that soybean exposed to stress factors such as drought and diseases resulted in a reduction in sugars, including glucose and fructose [35]. The ability of S genotypes to accumulate higher concentrations of raffinose and, especially stachyose, under NIR conditions compared to IR conditions, especially in 2013, may indicate the possible association of these two sugars with mechanisms of susceptibility, where raffinose and stachyose do not protect soybean plants from the damaging effects of *M. phaseolina*, especially under non-irrigated conditions, when there is more stress. It was previously reported that charcoal rot is more severe under drought conditions than under irrigated conditions [15]. Furthermore, previous research reported that the oligosaccharides raffinose and stachyose are involved in coping with stress responses to drought [9], seed desiccation [11], cold [10], reactive oxygen species [12], and sugar partitioning [13]. Maintaining higher levels of sucrose, glucose, and fructose in MR genotypes under IR and NIR in 2012 and 2013 indicated that MR genotypes had the ability to maintain high levels of these sugars under both charcoal rot and drought stress conditions. That is, susceptible genotypes responded to *M. phaseolina* by producing higher levels of raffinose and stachyose (oligosaccharides, RFOs) and lower levels of sucrose, glucose, and fructose. Conversely, the response of MR genotypes to *M. phaseolina* was to maintain high levels of sucrose, glucose and fructose, which could reflect the involvement of these sugars in the soybean’s mechanism of resistance against *M. phaseolina*. Changes in sugar levels in seeds of both S and MR genotype plants are consistent with these sugars playing a role in resistance to infection by the charcoal rot pathogen and possibly other biotic stress factors. Support for this conclusion is provided by the observations in S genotype plants of higher accumulations of raffinose and stachyose but lower accumulations of sucrose and glucose and by the observations in MR genotype plants of high sucrose, glucose, and fructose accumulations.

An isolate of *M. phaseolina* was shown to possesses genes coding for the production of peroxidases, oxidases, and hydrolytic enzymes that degrade polysaccharides and lignocelluloses in cell walls to penetrate the host [36]. Other enzymes involved in the pathogenesis caused by *M. phaseolina* include endoglucanase enzymes [37], amylases, proteases, hemicellulases, pectinases, and phosphatidases [38]. Several studies have reported that *M. phaseolina* produced phytotoxins with properties that suggest they could be used by the fungus to penetrate host tissue [39,40,41,42].

In addition, the *M. phaseolina* genome contains genes involved in purine biosynthesis, signal transduction, and carbohydrate esterases (CEs) [43] and membrane transporters, P450 s, transposases, glycosidases, and secondary metabolites [36]. The phytotoxins include asperlin, isoasperlin, phomalactone, phaseolinic acid, phomenon, and phaseolinone [35,41,42,44,45,46], as well as (-)-botryodiplodin [40,41,47]. On the other hand, host plants resistant to charcoal rot were shown to possess defense mechanisms of higher levels of antimicrobial compounds such as phenolics (phenols, lignin, and isoflavones) [48,49]. It was reported that lignin is a fundamental component of cell walls and seed coats, providing rigidity and structural support against pathogens [50,51]. The phytoalexin daidzein, one of the major isoflavones in soybean [52] was found to be involved in the defense mechanism against pathogens [33,52].

The effect of charcoal rot on sugars can also be explained in terms of the source–sink supply relationship in that charcoal rot limits the supply and form of exudates that include sucrose, glucose, fructose, and amino acids (glutamine, asparagine, alanine) [53,54,55]. Previous research has shown that the primary metabolites, including sucrose, glucose, and fructose, decline with plant development, whereas their storage forms, including raffinose, stachyose, and cell wall polysaccharides, increase with development [54,56,57]. We hypothesize that *M. phaseolina* affects the carbon–nitrogen pathway and sugar enzymes in susceptible genotypes, altering the availability of mono- and di-saccharides (sucrose, glucose, and fructose) and storage sugars (raffinose and stachyose). This leads to higher accumulation of raffinose and stachyose and lower sucrose, glucose, and fructose in susceptible genotypes (S) and higher sucrose, glucose, and fructose and lower raffinose and stachyose in moderately resistant genotypes (R) (Figure 2A–E). This proposed defense mechanism would be expected to operate along with, and in addition to, mechanisms contributing to charcoal rot resistance.

Our results show that the MR genotypes have the ability to maintain higher levels of sucrose, glucose, and fructose and lower levels of raffinose and stachyose under NIR, where charcoal rot was severe. This may indicate the possible involvement of these sugars in the defense mechanism of charcoal rot diseases. This conclusion cannot exclude the involvement of other mechanisms as reported above. The negative relationship between sucrose, fructose, and glucose vs. raffinose and stachyose may provide breeders with the potential to select for desirable sugars (sucrose, fructose, and glucose) for taste and flavor. 

### 4.2. Correlations between Sugars, and Sugar Frequencies

In 2012 and 2013, under both IR and NIR, stachyose levels positively correlated with raffinose levels and negatively correlated with sucrose, glucose, and fructose levels. Sucrose levels positively correlated with glucose or fructose levels (Table 6). Glucose levels positively correlated with fructose levels and negatively with raffinose or stachyose levels. Fructose levels also negatively correlated with. raffinose and stachyose levels (Table 6). Significant differences in the correlations between some sugar levels in different years could be due to hotter weather and lower precipitation in 2012 than in 2013 [19]. The substantial differences in sugar distribution and accumulation between genotypes under IR and NIR, particularly the high stachyose content under NIR conditions (Figure 3A–E), reflect the genotypic differences and may help explain the different sensitivity levels of MR or S genotypes to charcoal rot. Variation in the distributions of the various sugar levels suggests complexity in the relationships between the sugars and in their responses to irrigation and to charcoal rot disease.

## 5. Conclusions

Moderately resistant genotypes had lower levels of raffinose and stachyose compared to susceptible genotypes (Figure 2A–E). In this study, significant differences were observed in seed sugar levels between soybean genotypes with differing degrees of resistance to charcoal rot. The largest effects on seed sugar composition were observed under non-irrigated (NIR) conditions, including the following: (i) moderately resistant (MR) genotypes had elevated sucrose, glucose, and fructose levels; (ii) susceptible genotypes had elevated stachyose and lowered sucrose, glucose, and fructose levels; (iii) a negative correlation between stachyose and sucrose, glucose, and fructose levels; (iv) a negative correlation between raffinose and sucrose, glucose, and fructose levels; and (v) a positive correlation between sucrose and glucose and fructose levels. Raffinose and stachyose are among the types of secondary metabolites that were reported to be associated with increased resistance to infectious disease in plants [35]. However, in the current study the higher levels of raffinose and stachyose in S genotypes are associated with susceptibility and the higher levels of sucrose, glucose, and fructose in MR genotypes, and lower levels of raffinose and stachyose, are associated with resistance. This suggests that these sugars may play an important role as biochemical markers in the defense mechanism of soybean to charcoal rot. The higher levels of sucrose, glucose, and fructose in MR genotypes than in susceptible genotypes reflect the ability of MR genotypes to maintain the flux of di- and mono-saccharides from source (leaves) to sink (seeds) during the growth of the plants. Further research to include an assessment of sugars in a broader range of genotypes and to identify sugars at different stages of growth in both MR and S genotypes is needed to determine the flux and flow of mono- and di-saccharides from source to sink.

## Figures and Tables

**Figure 1 plants-12-00392-f001:**
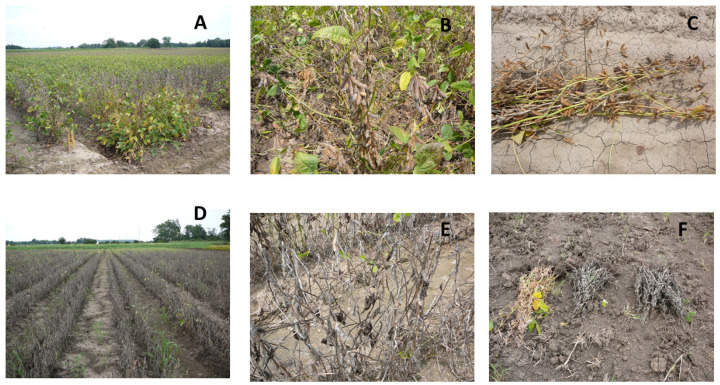
Charcoal rot of soybean in the field (**A**–**F**) under irrigated (**A**–**C**) and non-irrigated/drought (**D**–**F**). Notice the importance of irrigation management and the severity of charcoal rot under non-irrigation/drought conditions.

**Figure 2 plants-12-00392-f002:**
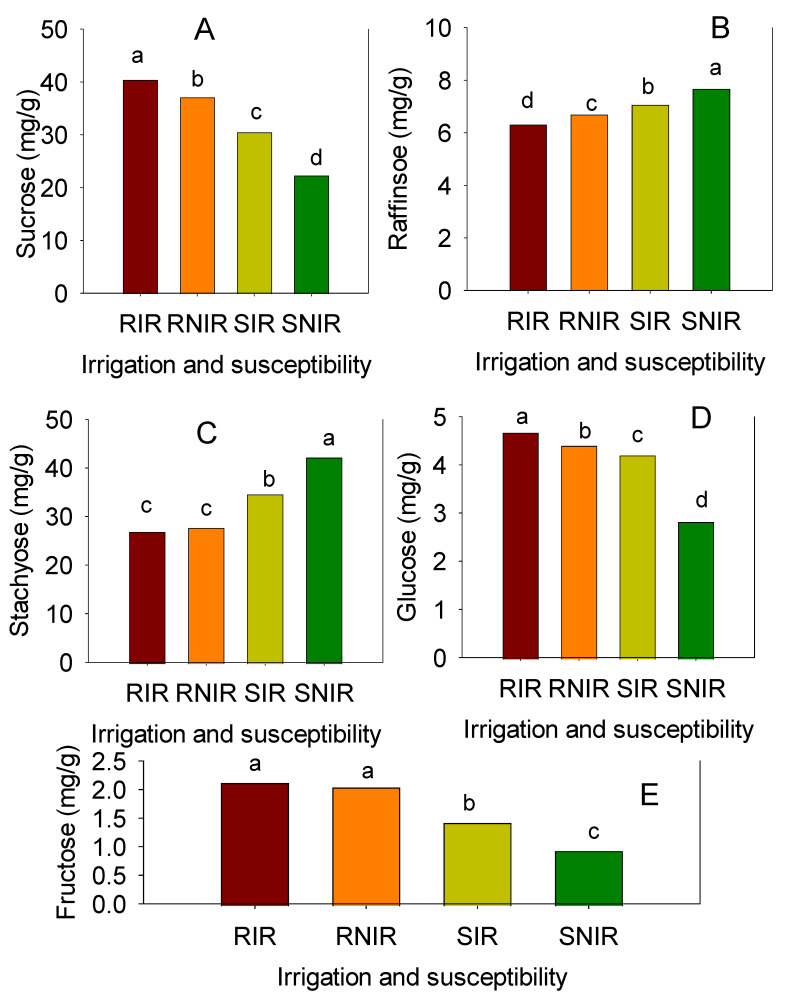
Soybean seed sucrose, (**A**); raffinose, (**B**); stachyose, (**C**); glucose, (**D**); and fructose, (**E**) across two years in moderately resistant (R) and susceptible (S) soybean genotypes under irrigated (IR) and non-irrigated (NIR). Moderately resistant genotypes (R) were seven: DS-880, DT97-4290, Dyna-Gro 36C44, Osage, R07-7232, USG 75Z38, USG Allen; and susceptible genotypes (S) were six: LS98-0358; Pharaoh; Progeny 4408; R01-581F; R02-1325; Trisoy 4788; two years were used; four replicates for each genotype were used. Letters that differ from each other in each column are significantly different at *p* < 0.05.

**Figure 3 plants-12-00392-f003:**
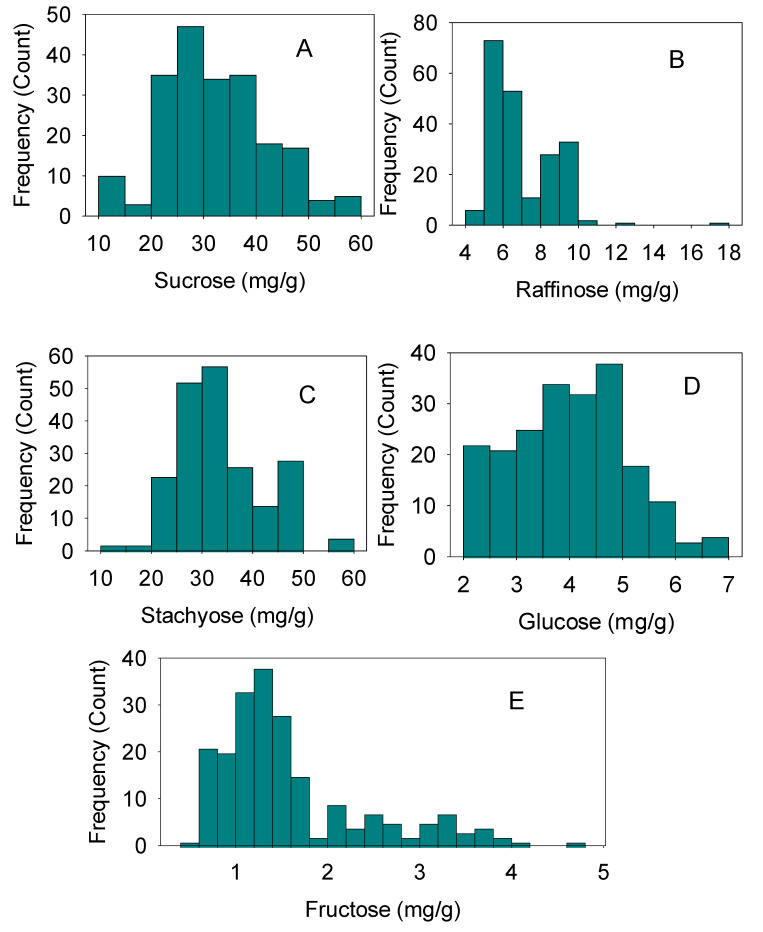
Distribution of soybean sugars (mg/g) across years, irrigation treatments, and genotypes. Sucrose, (**A**); raffinose, (**B**); stachyose, (**C**); glucose, (**D**); and fructose, (**E**). The experiment was conducted in 2012 and 2013 in Jackson, TN, USA. Gaps in *x*-axis in any distribution indicate there are no genotypes in that range. Frequency (*y*-axis) refers to the number of individual replicates of genotypes. Moderately resistant genotypes were seven: DS-880, DT97-4290, Dyna-Gro 36C44, Osage, R07-7232, USG 75Z38, USG Allen; and susceptible genotypes were six: LS98-0358; Pharaoh; Progeny 4408; R01-581F; R02-1325; Trisoy 4788; two years were used; four replicates for each genotype were used.

**Table 1 plants-12-00392-t001:** Charcoal rot classifications (MR and S) of the thirteen soybean genotypes included in this study, of which six are maturity group (MG) IV and seven are MG V. This table was modified from Mengistu et al. (2018).

Genotype	Maturity Group	Resistance/Susceptibility to Charcoal Rot
DS-880	MG V	Moderately Resistant
DT97-4290	MG IV	Moderately Resistant
R07-7232	MG V	Moderately Resistant
USG 75Z38	MG V	Moderately Resistant
USG Allen	MG V	Moderately Resistant
Osage	MG V	Moderately Resistant
Dyna-Gro 36C44	MG IV	Susceptible
Progeny 4408	MG IV	Susceptible
R01-581F	MG V	Susceptible
R02-1325	MG V	Susceptible
Trisoy 4788	MG IV	Susceptible
LS98-0358	MG IV	Susceptible
Pharaoh	MG IV	Susceptible

**Table 2 plants-12-00392-t002:** *Macrophomina phaseolina* infection levels (colony forming units, CFU/g) determined at growth stage R7 under irrigated and non-irrigated conditions for the thirteen soybean genotypes in 2012 and 2013. Table was modified from Mengistu et al. (2018).

	2012					2013		
Genotype	Irrigated			Non-Irrigated		Irrigated		Non-Irrigated
DS-880	1572	cd *	722	gf	1409	e	3175	dc
DT97-4290	212	e	590	g	1350	e	1278	d
R07-7232 (R07)	1364	cd	1818	egf	1976	e	5731	c
USG 75Z38 (USG75)	1158	cde	2459	edf	1643	e	3062	dc
USG Allen (USGAl)	567	ed	986	gf	2257	ed	3826	c
Osage	2249	bcd	4787	edc	1953	e	3833	c
Dyna-Gro 36C44 (Dayna)	11,427	ab	39,704	ba	7555	bc	43,326	a
Progeny 4408 (P4408)	1171	cde	10,770	bc	6206	bcd	24,428	ba
R01-581F (R01)	5195	abc	10,431	bc	25,879	a	30,004	ba
R02-1325 (R02)	3634	abc	12,694	bac	3491	ecd	39,340	a
Trisoy 4788 (T4)	3104	bcd	9438	dc	3730	ecd	15,142	b
LS98-0358 (LS98)	6635	abc	37,097	ba	15,528	ba	42,328	a
Pharaoh	18,905	a	43,547	a	17,633	ba	32,183	ba

* Letters that differ from each other in each column are significantly different at *p* ≤ 0.05. Abbreviations of genotypes: DS-880 = DS-880; DT97-4290 = DT97-4290; R07-7232 = R07; USG 75Z38 = USG75; USG Allen= USGAl; Osage = Osage; Dyna-Gro 36C44 = Dayna; Progeny 4408 = P4408; R01-581F = R01; R02-1325 = R02; Trisoy 4788 = T4; LS98-0358 = LS98; Pharaoh = Pharaoh.

**Table 3 plants-12-00392-t003:** Analysis of variance (ANOVA) of the effects of the main factors (year, genotype, irrigation), and their interactions, on seed sugars (mg/g) (sucrose, raffinose, stachyose, glucose, and fructose) among 13 soybean genotypes. The experiment was conducted in 2012 and 2013 in Jackson, TN, USA.

		Sucrose		Raffinose		Stachyose		Glucose		Fructose	
Effect	DF	F	*p*	F	*p*	F	*p*	F	*p*	F	*p*
Year (Y)	1	44.17	***	72.34	***	0.48	ns	0.06	ns	190.57	***
Genotype (G)	12	12.59	**	2.97	ns	16.43	***	6.32	***	23.93	***
Y*G	12	4.18	ns	2.18	ns	1.38	0.18	5.27	**	12.30	***
Irrigation (IR)	1	32.70	***	4.49	ns	28.94	***	42.45	***	26.78	***
Y*IR	1	0.46	ns	1.46	ns	6.02	*	9.50	**	2.26	ns
G*IR	12	0.79	ns	0.31	ns	1.53	ns	2.31	ns	1.27	ns
Y*G*IR	12	0.16	ns	0.27	ns	0.99	ns	0.59	ns	0.72	ns
Residuals		28.51		0.18		35.94		0.38		0.15	

* Significance at *p* ≤ 0.05; ** significance at *p* ≤ 0.01; *** significance at *p* ≤ 0.001; ns, not significant.

**Table 4 plants-12-00392-t004:** Levels (mg/g) of seed sucrose (Suc), raffinose (Raff), stachyose (Stac), glucose (Glu), and fructose (Fru) in moderately resistant (MR) and susceptible (S) soybean genotypes to charcoal rot infection under irrigated (IR) and non-irrigated (NIR) conditions. The experiment was conducted in 2012 in Jackson, TN, USA.

					IR	2012										
Genotype	DS- 880	DT97-4290	Dyna	Osage	R07	USG75	USGAl	MR Mean	LS98	Pharaoh	P4408	R01	R02	T4	S Mean	LSD *
Resistance	MR	MR	MR	MR	MR	MR	MR		S	S	S	S	S	S		
Sugars																
Suc	37.85	30.38	31.50	30.70	39.83	29.23	37.73	33.89	30.58	24.85	32.78	35.38	27.68	27.30	29.76	1.90
Raff	5.27	5.25	6.29	5.48	5.55	5.50	5.55	5.56	6.88	6.13	6.13	5.75	6.28	6.55	6.29	0.31
Stac	25.38	25.98	42.10	26.53	28.80	25.28	27.75	28.83	40.88	37.15	31.90	33.75	37.20	31.88	35.46	2.50
Glu	3.92	4.21	4.28	4.73	4.50	4.09	4.81	4.36	4.70	4.48	5.62	4.82	5.00	4.71	4.89	0.35
Fru	1.64	1.48	1.44	1.56	1.56	1.47	1.55	1.53	1.35	1.26	1.21	1.15	1.28	1.40	1.28	0.07
					NIR	2012										
Genotype	DS-880	DT97-4290	Dyna	Osage	R07	USG75	USGAl	MR mean	LS98	Pharaoh	P4408	R01	R02	T4	S mean	LSD *
Resistance	MR	MR	MR	MR	MR	MR	MR		S	S	S	S	S	S		
Sugars																
Suc	31.85	27.25	19.95	28.00	36.38	24.60	34.73	28.97	20.80	17.53	24.05	19.60	21.30	22.30	20.93	2.87
Raff	5.50	5.57	6.44	5.75	5.85	5.73	5.07	5.70	6.15	6.30	6.42	6.61	6.65	7.33	6.58	0.44
Stac	29.08	28.73	45.43	27.70	29.70	27.30	26.35	30.61	45.85	40.50	33.00	34.85	42.65	35.05	38.65	2.67
Glu	3.43	3.86	2.53	3.92	3.92	3.55	4.48	3.67	2.90	2.61	2.82	2.73	2.46	4.88	3.07	0.30
Fru	1.35	1.21	1.03	1.25	1.24	1.13	1.16	1.20	0.84	0.93	0.73	0.78	0.93	0.79	0.83	0.08

* LSD, least significant difference test; significant at *p* ≤ 0.05 within each row. The difference between the values of any two genotypes is statistically significant if it equals or exceeds the corresponding LSD. MR mean = mean across all MR genotypes; S mean = mean across all S genotypes. Abbreviations of genotypes: DS-880 = DS-880; DT97-4290 = DT97-4290; R07-7232 = R07; USG 75Z38 = USG75; USG Allen= USGAl; Osage = Osage; Dyna-Gro 36C44 = Dayna; Progeny 4408 = P4408; R01-581F= R01; R02-1325 = R02; Trisoy 4788 = T4; LS98-0358 = LS98; Pharaoh= Pharaoh.

**Table 5 plants-12-00392-t005:** Levels (mg/g) of seed sucrose (Suc), raffinose (Raff), stachyose (Stac), glucose (Glu), and fructose (Fru) in moderately resistant (MR) and susceptible (S) soybean genotypes to charcoal rot infection under irrigated (IR) and non-irrigated (NIR) conditions. The experiment was conducted in 2013 in Jackson, TN, USA.

					IR	2013										
Genotype	DS-880	DT97-4290	Dyna	Osage	R07	USG75	USGAl	MR Mean	LS98	Pharaoh	P4408	R01	R02	T4	S Mean	LSD *
Resistance	MR	MR	MR	MR	MR	MR	MR		S	S	S	S	S	S		
Sugars																
Suc	34.73	50.55	35.23	47.55	39.70	45.30	46.70	42.82	27.75	28.90	29.85	31.65	31.35	31.63	30.19	2.91
Raff	5.07	8.02	9.07	7.42	6.99	6.59	6.30	7.07	9.37	8.82	9.29	6.38	6.05	5.85	7.63	0.52
Stac	26.35	26.53	36.95	28.00	28.20	27.45	28.33	28.83	32.65	32.13	31.25	33.18	30.70	34.25	32.36	3.32
Glu	4.48	4.58	3.62	4.83	4.38	6.32	4.64	4.69	3.93	3.73	3.93	3.76	3.22	3.12	3.62	0.35
Fru	1.16	2.75	1.27	3.42	2.59	2.21	2.26	2.24	1.37	1.46	1.15	1.89	1.69	2.22	1.63	0.30
					NIR	2013										
Genotype	DS-880	DT97-4290	Dyna	Osage	R07	USG75	USGAl	MR mean	LS98	Pharaoh	P4408	R01	R02	T4	S mean	LSD *
Resistance	MR	MR	MR	MR	MR	MR	MR		S	S	S	S	S	S		
Sugars																
Suc	47.98	46.43	27.30	46.28	37.95	42.05	41.53	41.36	23.40	22.00	24.60	20.13	22.18	27.63	23.32	2.66
Raff	8.54	7.95	11.33	8.26	8.03	6.99	7.09	8.31	9.85	8.92	9.41	6.25	8.66	7.07	8.36	0.85
Stac	25.28	25.23	45.53	28.70	29.50	26.20	30.13	30.08	42.40	46.98	43.70	44.00	45.30	47.28	44.94	2.98
Glu	4.53	5.15	2.83	4.63	4.51	6.06	4.84	4.65	2.91	2.36	2.84	2.48	2.84	2.46	2.65	0.23
Fru	2.79	3.33	0.92	3.51	2.80	2.61	2.41	2.62	0.89	1.16	0.84	1.14	0.99	1.38	1.07	0.25

* LSD, least significant difference test; significant at *p* ≤ 0.05 within each row. The difference between the values of any two genotypes is statistically significant if it equals or exceeds the corresponding LSD. MR mean = mean across all MR genotypes; S mean = mean across all S genotypes. Abbreviations of genotypes: DS-880 = DS-880; DT97-4290 = DT97-4290; R07-7232 = R07; USG 75Z38 = USG75; USG Allen = USGAl; Osage = Osage; Dyna-Gro 36C44 = Dayna; Progeny 4408 = P4408; R01-581F = R01; R02-1325 = R02; Trisoy 4788 = T4; LS98-0358 = LS98; Pharaoh = Pharaoh.

**Table 6 plants-12-00392-t006:** Pearson correlation coefficients (R) and their probability (*p*) between seed sugars (sucrose (Suc, mg/g), raffinose (Raff, mg/g), stachyose (Stac, mg/g), glucose (Glu, mg/g), and fructose (Fru, mg/g)) for 2012 and 2013 across soybean genotypes and within irrigation treatments. The experiment was conducted in 2012 in Jackson, TN, USA.

		2012	IR				2012	NIR	
	Suc	Raff	Stac	Glu		Suc	Raff	Stac	Glu
Raff	R = −0.27				Raff	R = −0.33			
	*p* = *					*p* = **			
Stac	R = ns	0.40			Stac	R = −0.39	0.31		
	*p* = ns	***				*p* = ***	*		
Glu	R = ns	ns	ns		Glu	R = 0.40	ns	−0.36	
	*p* = ns	ns	ns			*p* = ***	ns	**	
Fru	R = ns	ns	−0.41	−0.27	Fru	R = 0.42	−0.42	−0.41	ns
	*p* = ns	ns	***	*		*p* = ***	***	**	ns
		2013	IR				2013	NIR	
	Suc	Raff	Stac	Glu		Suc	Raff	Stac	Glu
Raff	R = ns				Raff	ns			
	*p* = ns					ns			
Stac	R = ns	ns			Stac	−0.76	ns		
	*p* = ns	ns				***	ns		
Glu	R = 0.56	ns	ns		Glu	0.74	−0.29	−0.78662	
	*p* = ***	ns	ns			***	*	***	
Fru	R = 0.52	−0.30	−0.41	0.41	Fru	0.79	ns	−0.72	0.73
	*p* = ***	*	**	**		***	ns	***	***

* Significance at *p* ≤ 0.05; ** significance at *p* ≤ 0.01; *** significance at *p* ≤ 0.001; ns, not significant. Moderately resistant genotypes (MR) were seven: DS-880, DT97-4290, Dyna-Gro 36C44, Osage, R07-7232, USG 75Z38, USG Allen; and susceptible genotypes (S) were six: LS98-0358; Pharaoh; Progeny 4408; R01-581F; R02-1325; Trisoy 4788; two years were used; four replicates for each genotype were used.

## Data Availability

Not applicable.

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
