# Peer review of "Soybean Seed Sugars: A Role in the Mechanism of Resistance to Charcoal Rot and Potential Use as Biomarkers in Selection"

_plants, 2023, doi:10.3390/plants12020392_

Round 1

Reviewer 1 Report

This work seems interesting, and its authors lead research in this field, which is an indication that this research could be of quality.

My biggest concern about this work is that it has been working with aged seeds. Sucrose is easily hydrolyzed to glucose and fructose, in addition, all these sugars are easily isomerizable, and the relative content of sugars could have changed since the harvest (2012-2013).

Therefore, the results could be biased due to this fact, and the conclusions of this work could be erroneous. Sugars should have been analyzed immediately after seed collection.

In this regard, it has been cited that the Maillard reaction in soy seeds may be a consequence of the formation of reducing sugars through a gradual hydrolysis of oligosaccharides during aging.

(Sun, W. Q., & Leopold, A. C. (1995). The Maillard reaction and oxidative stress during aging of soybean seeds. Physiologia Plantarum, 94(1), 94-104.)

Therefore, the authors should argue about this fact.

Reviewer 2 Report

I've reviewed the paper "Soybean Seed Sugars: a Role in the Mechanism of Resistance to Charcoal Rot and Potential Use as Biomarkers in Selection. The author investigated the association between seed sugars and their role as biomarkers in the soybean defense mechanism in the moderately resistant and susceptible genotypes to charcoal rot. I believe that this study provides valuable information on charcoal rot. Although there are only two years of data in this paper, I do not request it at this time because little is known about the findings on charcoal rot. There are several questions that need to be answered and corrected below.

1)    Tables 3 and 4 are only different years; please combine them into one table.

2)    Tables 6 and 7 also differ only in year, so please put them in one table.

3)    L191-198 Correlation of sugars should be described correctly.

4)    Please explain the results of Figure 2 in the Results paragraph, not in the Discussion. There is no error bar. Please indicate the number of repetitions (n=). Please describe the specific genotypes used.

5)    Please describe the specific genotype used in Figure 3.

It is possible that Tables and Figure are just separate charts from the original data; if Figures 2 and 3 are the same as the data used in the Table, this paper should be rejected.

Round 2

Reviewer 1 Report

The authors have satisfactorily responded to the concerns raised, so I believe that this manuscript is appropriate to be published in its current form.